# A Novel Educational Smartphone Application for Cognitively Healthy Seniors: A Pilot Study

**DOI:** 10.3390/ijerph18126601

**Published:** 2021-06-19

**Authors:** Blanka Klimova, Lukas Sanda

**Affiliations:** Department of Applied Linguistics, Faculty of Informatics and Management, University of Hradec Kralove, Rokitanskeho 62, 500 03 Hradec Králové, Czech Republic; lukas.sanda@uhk.cz

**Keywords:** older people, smartphone application, English, learning, usability testing

## Abstract

Modern technologies surround people every day, including seniors. The aim of this pilot study was to create a maximally user-friendly mobile application in order to meet older users’ individual needs. The research sample consisted of 13 older individuals at the age of 55+ years with a mean age of 67 years, living in the Czech Republic. The key assessment tools of this pilot study were the developed application and usability testing. The findings confirmed that the newly developed mobile application for teaching English met the needs of cognitively healthy seniors, and was acceptable and feasible. In addition, it indicated what technical (e.g., visual interface or easy navigation) and pedagogical (e.g., an instructional manual or adjusting to seniors’ learning pace or clear instructions) aspects should be strictly followed when designing such an educational smartphone application. In addition, the authors of this pilot study provide several implications for pedagogical practice. Further research should include more empirical studies aimed at the exploration of educational mobile applications for older generation groups with respect to meeting their individual needs in order to enhance their overall well-being. However, such studies are, nowadays, very rare.

## 1. Introduction

Many years have passed since the Internet became available to the public. Today, it is more of a rarity that the household does not have at least one personal computer or laptop. It is used to store memories in the form of photos and videos, communicate between family and friends, work or just to spend free time watching movies, listening to music or playing computer games. Modern technologies surround people every day, including seniors.

On average, older adults spend around 27 h online per week [1]. Most frequently, they use search engines to find information on topics of their interest, such as checking weather, reading news, or shopping discounts. Furthermore, 70% of older people aged between 55 and 64 years and 53% seniors at the age of 65–74 years own a mobile phone, of whom only 59% own a smartphone [1,2]. For people over the age of 80, only 17% of them have a smart mobile phone [2]. The reasons for the decline of the use of both mobile phones and smartphones among older seniors might be financial limitations, vision and hearing impairments, a lack of interest and knowledge in using technological devices and their advanced functionalities [3], as well as lower abilities to use these technological devices [4]. These findings were also confirmed by Briede-Westermeyer et al. [5] who reported that the use of mobile phones was greater among the younger elderly (55 to 69), who possess a higher educational level, higher incomes, and who perceived having better health than older seniors.

By contrast, one of the most recent research studies by Busch et al. [6] conducted among Norwegian seniors at the age of 60+ years reveals that there was a low prevalence of problematic smartphone use among these older people. The findings also indicate that seniors used smartphones for various social and non-social reasons and that social media and news reading were the most frequent reasons for their use. In addition, the findings of a Canadian research study [7] show that retired older people use mobile devices, such as a mobile phone or a tablet, to access not only their social networks, but also to enrich and expand their informal, self-directed learning. In fact, the authors of this study discovered that 78.5% of their respondents at the age of 55+ years agreed or strongly agreed with the statement that the availability of these mobile devices made their learning easier. Furthermore, 46.8% of the respondents admitted using mobile applications to support their mobile learning. Thus, it seems that younger older generation between 55 years and 74 years appear to be able to use these mobile devices for learning purposes in comparison with older seniors at the age of 75+ years.

## 2. Literature Review

Nowadays, if seniors at the age of 55+ years want to use these mobile devices to study foreign languages, the market is not very friendly. There are several mobile applications, such as Duolingo, EWA: English, Mondly: Learn 33 languages, or LinGo: Play, aimed at learning English [8,9,10]. However, none of them is targeted at seniors in a sense that reflects their individual needs, such as possessing a minimalist design, avoid redundant and irrelevant content, have clear instructions on how to use the application, have simple and easy navigation in the application, and avoid complex controls [11], which is a pity since these educational applications, apart from enabling seniors to stay socially inclusive, can improve their working memory and reasoning skills [12]. Migo et al. [13] also claim that mobile apps can serve as an external memory aid. In addition, the older people appreciate having instructions in their native language.

Thus, generally there are very few empirical studies on the exploration of the use of mobile applications for learning English as a foreign language by older adults [14,15,16]. However, the technology and in this case, smartphones can partly contribute to the solution of the issue of aging population [17]. Internet literacy can increase older people’s level of participation in economy and society, and overall, improve their quality of life and well-being [18]. Hong et al. [18] claim that the use of smartphones does significantly impact Internet literacy and its use by older individuals. Moreover, research provides conclusive evidence that the use of mobile devices helps seniors with their self-realization, psyche, socializing, and self-confidence [16,19]. Furthermore, the results of a few empirical studies on computer-based foreign language learning programs, e.g., [20,21,22], reveal that there might be some cognitive benefits for healthy elderly people, especially the enhancement of their cognitive skills, such as working memory.

As far as the mobile applications for learning English are concerned, Wang and Christiansen [15] conducted an empirical study among 55 older Chinese individuals aged between 40 and 81 years. The results of this study show that the participants improved their vocabulary, as well as persisting in learning for 17 weeks since they were intrinsically motivated to learn English. The same was true for the study by Ronda and Belda-Medina [23] who performed an empirical study with 60 Spanish seniors at the age 50+ years at the University of Alicante. They ran a 14-week English Communicative Workshop; each week students had three lessons. The findings of this study show that at the end of the course seniors improved their vocabulary in English. Apart from that, their self-confidence in English increased and they were less afraid to talk in class in front of their peers. Students also reported to know better their classmates thanks to the use of the Whatsapp group in English and to socialize and feel more motivated to learn about English and information and communication technologies (ICTs) in general.

The aim of this pilot study, therefore, was to create a maximally user-friendly mobile application in order to meet older users’ individual needs. The research question was set as follows: does the newly developed smartphone mobile application for teaching English meet the needs of cognitively healthy seniors?

## 3. Materials and Methods

### 3.1. Research Sample

Originally, the research sample consisted of 17 older individuals at the age of 55+ years with a mean age of 67 years. However, four participants dropped during the first week of conducting the study, and only 13 participants then remained. Nine participants were females and five were males. Out of these, nine participants attained secondary education and four had a university degree. Although the sample seems to be small, the usability testing requires a minimum of five users only [24]. All seniors were Czech citizens and all of them were students of the University of the Third Age (U3V) at the University of Hradec Kralove, Czech Republic. Their level of English was at A2 (pre-intermediate) level according to the standardized Common European Framework of Reference for Languages (CEFR). The participants did not suffer from any serious cognitive or mental impairments, such as dementia. The participants also had to possess a smartphone with the Android operating system version 6 and higher. They all knew how to work with the basic functions of their smartphone, such as making calls, photos, sending e-mails, texting, or using a calculator. In order to eliminate cognitively impaired seniors, all participants underwent the Czech adaptation of the Montreal Cognitive Assessment (MoCA), specifically the version of MoCA-CZ1 [25,26] to assess cognitive abilities.

The research complied with the Helsinki declaration because all participants were informed about the purpose of the study, and about the possibility of withdrawing from participation in the research at any time, as well as giving informed and voluntary consent to participate in this research. The study was also confirmed by the Committee for Research Ethics at the University of Hradec Králové, no. 2/2021.

### 3.2. Research Tools and Methods

The participants were offered to test a newly developed mobile application with one model lesson based on the textbook Angličtina pro seniory (English for Seniors) corresponding to their level of English [27], as well as to their other individual needs. The content of the mobile application was consulted with an English teacher, one of the authors of this article. The mobile application was then developed by another author of this study, a programmer. The purpose of the application was to create a product that will be available during an English language course at U3V of the University of Hradec Králové, as a practice tool for course participants outside the face-to-face classes. The testing was performed during March 2021 and the current version of the mobile application was accessible to them through the Firebase platform.

The application consists of one model lesson offering various learning modules—reading, writing, listening, vocabulary, phrases, grammar, practicing vocabulary and general exercises. Most modules, in addition to writing, grammar, vocabulary and phrases, contain questions that the user answers. There are three ways to answer—plain text, yes/no answers, and audio recording. The writing module does not contain questions, but one text field, where there is space for writing answers from a specific user. No response from the user is required. This means that the user can continue the lesson without answering one of these answers. It is possible to return to each module or to the entire lesson at any time and add missing information/answers or edit existing answers. There are no questions for grammar, words and phrases, as they are purely informative modules. When completing a lesson, in the sense of at least going through all the modules, it is possible to access other lessons that are available. If there are a total of 13 lessons on offer, it is not possible to complete Lesson 1 and then freely skip to Lesson 11. It is necessary to complete the lessons in their chronological order, which is required from the point of language acquisition (Figure 1).

Each lesson, in addition to its teaching modules, also contains tasks and tests. Each task has a fixed date by which it should be completed. In addition, there is a time limit (15 min) for tests, during which the test should be completed. After its expiration, the results provided by the user will be sent automatically and the test is terminated. At the end of each test, the status of the correct answers to the total number of questions is displayed. The user can then return to the test to view his/her answers against the correct answers listed. As the target group is seniors, the application offers a clear user interface in which they can set the font size in the application. This setting takes place during the introductory tutorial, which can also be adjusted later in the application settings. In addition, the testers were sent a user manual, as well as a video tutorial on how to upload and use the mobile application. The participants were also offered the ability to contact the developer of the application by phone.

Overall, the programmer of the app considered seniors’ aging problems, such as worsen eyesight or slower pace and attempted to develop a modifiable text size, minimalist design, relevant content without superfluous elements, as well as to keep the content clear, simple and user-friendly.

Simultaneously with the mobile application, a web application was created. It serves the teacher as an environment for adding new members of the course, new content to the mobile application, i.e., new lessons, tasks and tests, and for the ability to control responses from users from the mobile device, as well as to send them feedback on their performance.

Another assessment tool was usability testing. This is a technique used in user-centered interaction design to evaluate a product by testing it on users. It focuses on the design intuitiveness of the product and it is tested with users who have no prior exposure to it. Such testing is key to the success of an end product [28]. Usability testing was mediated by a mobile application, as well as by a method of focus group interviews, which assumes a smaller number of people who are asked specific questions and their answers form the test result itself. Unfortunately, due to the COVID-19 pandemic, the interview with individual persons could not be interviewed face-to-face and the authors of this study had to contact them online by sending them a questionnaire to discover the answers to the set research questions. This questionnaire contains 35 questions, out of which 21 were closed questions and 14 were open questions, related to the use of mobile applications for learning English as a second language. However, due to the technical problems (i.e., some of the seniors were not able to upload the application) or senior’s disinterest in using the mobile app, only 76.47% (*n* = 13) of 17 addressed seniors tested the app and completed the questionnaire. The questionnaire included the questions related to the following areas: demographic information, mobile application and its aspects, learning modules, tests and tasks, evaluation of the application as a whole, and an additional section for further inquiries that could not be assigned to the questions in other sections.

## 4. Results

The results of the ***demographic data*** area reveal that there were 13 participants aged between 57 years and 75 years, and with female prevalence (69%). This is not surprising finding since research [29,30] shows that females are more likely to become involved in online activity characterized by communication and exchanging of information whereas males are more likely to engage in online activity that focuses on searching for information. In addition, the majority of the participants (69%) had secondary education.

As far as the ***piloted mobile application and its aspects*** are concerned, there were 10 questions altogether. Since this is a mobile application, where it is not customary to have a user manual and/or a video tutorial, the first question concerned the fact whether the respondents/testers would appreciate having them before the initial launch of the application, which was also the case in this pilot study. The seniors did receive both the user manual and the video tutorial how to upload the developed mobile application on their smartphone. Figure 2 below illustrates that 61.5% (*n* = 8) of respondents agreed that these were useful elements and another 30.8% (*n* = 4) answered “Maybe”, which means that this manual and/or the video tutorial on how to upload the application and use it is relevant.

Two questions concerned the visual aspects of the mobile application, with which the respondents were satisfied and found it appealing. As for the control of the application, 38.5% of the testers reported that some elements were not completely meaningful in themselves and they were not understood until after a longer time of using the application (Figure 3). There was even one answer stating that the control element/button is a bit confusing. However, this answer, on the basis of the examination of the other answers, indicates that the respondent would rather appreciate this application on a device with larger dimensions, specifically on a personal computer or laptop.

In addition, the majority of the respondents (84.6%/11) confirmed that the mobile application by its functions and overall appearance corresponds to the needs of older generation and meets their expectations. 30.8% (*n* = 4) of the respondents stated that it did not suit them due to the size of the device, which appeared to be the main drawback associated with the smaller size of the smartphone.

Interestingly, the testers reported that this mobile application could be exploited not only as a tool for practicing outside the traditional, face-to-face classes, but also directly during the course. 61.5% (*n* = 8) stated that this application could also replace physical learning materials and another 15.4% (*n* = 2) would not mind.

In the follow-up question, whether they would like to use this application during their studies of English at U3V, 76.9% (*n* = 10) said that they would like to use it, but not as a primary learning tool. However, the remaining 23.1% (*n* = 3) were of the opinion that such an application could even replace all teaching materials and become a key element in their foreign language learning (Figure 4). Generally, the participants found the application playful, brisk, and entertaining.

The key element of the ***learning modules*** (reading, writing, listening, vocabulary, phrases, grammar, practicing vocabulary and general exercises), through which all learning, practice, and tests are mediated, met seniors’ expectations. There was only one answer stating that the modules did not meet expectations, which concerned a more detailed description of the lesson (Figure 5). However, this was most likely connected with the content that was taken over from the textbook [25] some of which does not seem to be suitable for the smartphone application. Generally, more than half of the respondents were satisfied with each module. The only thing the respondents missed was a dictation. Other questions in this describe seniors’ satisfaction with individual modules, out of which they especially appreciated the writing module, which contained an example on how to start with a written answer; 84.6% (*n* = 11) of the respondents praised this highly. Based on all the answers, the listening module seemed to be very popular, as well as the recorded pronunciation of new words and phrases, which the participants were supposed to learn. Nevertheless, they (46.2%/*n* = 6) would be grateful if they could pause the listening and then continue further on, not just interrupt it. They would also welcome a slower pace of speaking on the recording. In fact, this approach should also be considered in classical teaching, i.e., adapting teacher’s pace to the pace of seniors.

What was the respondents’ view on the possible existence of an application for practicing during the face-to-face U3V English courses using a mobile application? According to Figure 6 below on scale 1–5 where 1 is considered to be excellent, 2—very good, 3—good, 4—satisfactory and 5—unsatisfactory, 61.5% (*n* = 8) of the respondents rated this idea as excellent and 30.8% (*n* = 4) very good.

The findings of the final ***evaluation of the application as a whole*** revealed that the idea to implement a mobile application in foreign language learning was a very good choice. Figure 7 below illustrates that 92.3% (*n* = 12) of the respondents welcomed this idea and they considered it excellent or very good. Only one respondent reported that it was satisfactory, which was associated with his/her preference for a bigger size of the device.

Figure 8 then demonstrates seniors’ overall evaluation of the control of the mobile application, which indicates its easy use and friendliness as the results reveal since 84.6% (*n* = 11) consider its use/control as excellent or very good. Only one respondent again considered it satisfactory, which was once again connected with the absence of its version for a desktop device.

The results of the final evaluation of the learning modules and tasks and tests ranged from excellent to good with the prevalence of the excellent evaluation (61.5%/*n* = 8) as Figure 9 below illustrates.

Overall, the findings show that 92.3% (*n* = 12) of the participants provided a very positive evaluation and only one answer was only “good” (Figure 10).

In the last section of the questionnaire there was space for ***additional seniors’ comments/enquiries***. There was a total of five answers, which reflected the same opinion, i.e., the size of the mobile phone was not sufficient and it would be appropriate to develop an application for a personal computer. This comment is obviously important; however, the purpose of this pilot study was to develop a maximally friendly educational application for mobile devices. A mobile phone eliminates the need to carry a laptop. The average weight of a laptop is between 2 and 2.5 kg with an average display size of 13 (33 cm) to 15.6 (39.6 cm) inches. For this reason, the mobile phone/smartphone appears to be an ideal device that does not take up so much space and is not difficult to carry. Therefore, it is possible to practice a foreign language on a mobile device anywhere and at any time. Figure 11 below then provide a summary of the key findings.

## 5. Discussion

The findings of this pilot study clearly show that seniors, specifically those with a mean age of 67, are able to use a smartphone and its applications for self-learning purposes. This is in line with recent research studies on this topic [6,7]. In fact, Toyota et al. [31] performed a study with a sample of 10 subjects whose mean age was the same as in this pilot study and found out that their novice senior smartphone users were able to use an interactive self-learning mobile application in order to learn how to do several basic smartphone operations and how to use a typical map application. This finding still contradicts a prevailing opinion of the majority studies, which state that seniors have negative attitudes towards the use of mobile phones as a teaching aid, although they used them for conducting everyday tasks [32,33]. However, the educational mobile application that is tailored to the seniors’ needs might generate a lot of benefits to them, such as increasing older people’s confidence about using smartphones, enhancing intergenerational communication, developing informal learning, or practicing their cognitive skills [15,16,19,20,21,22,34].

The results of this pilot study then indicated what specific features this *ideal* smartphone application should possess. The findings revealed that it is desirable to accompany it with an instructional manual or a video in order to provide seniors with information about its uploading and use. Leung et al. [35] also confirm this as their elderly participants preferred using the instruction manual. Furthermore, Liou [36] claim that technical support is necessary in order to enable seniors an easier use of their smartphone application. He suggests providing an introduction page showing the usefulness of a mobile app and how it works. He also maintains that the face-to-face instruction can be more useful to seniors who have low self-efficacy on technology.

In addition, the findings of this pilot study illustrated that the majority of the respondents (84.6%/11) reported that the smartphone application by its functions and overall appearance corresponded to the needs of older generation and met their expectations. The reason is that it has simple and easy navigation and it is visually appealing as the respondents confirmed. Moreover, the text size is sufficient (18–20 pixels); the icons and controls seem to be big enough; sans serif font, which is easy to read, has been implemented; the same font is used across the whole applications; and there is a high contrast between the background and the text itself. Furthermore, research shows that one of the most important factors is the customization of the visual side of the application because with age, the eyesight of almost every individual gradually deteriorates [37,38]. This is completely different from younger learners who do not need this that much and have different requirements, such as the higher speed of interactions or sharing videos [39].

The majority of the participants 92.3% (*n* = 12) were especially positive about the learning modules (reading, writing, listening, vocabulary, phrases, grammar, practicing vocabulary and general exercises). There were, of course, some smaller insufficiencies, such as a requirement to pause the listening or more detailed instructions in some modules. However, it seems that the content of the application was meaningful and clearly met seniors’ expectations and needs. Thus, it can be expanded into additional mobile lessons for learning English. This finding provides evidence for the fact that the newly designed application can be used for educational purposes since it fully satisfies seniors’ individual needs. In fact, the participants also admitted exploiting this application not only outside the traditional, face-to-face classes, but also directly during their U3V courses.

The only drawback of this application is that the size of a smartphone is still relatively small for seniors. As one of them points out: 


*Working on tasks on a larger screen (tablet, computer) is more enjoyable for me. However, I consider practicing lessons using a smartphone very beneficial because we do not always have the time and space to work with the above devices. We basically always have a mobile phone with us and the possibility of practicing is much more accessible.*


Overall, based on the results of this pilot study, the newly designed smartphone application, its development and usability testing, have several implications for pedagogical practice. Firstly, all self-learning, such as learning a foreign language via a smartphone application, must be accompanied by a manual, preferably by a video tutorial where all the learning stages, including the lesson aims, as well as technical functionalities are described. Secondly, all tasks must be provided with clear and detailed instructions, written in participant’s native language and have a consistent design. Thirdly, a teacher must adapt to his/her learner’s pace in order not to discourage them from their studies, which is not that obvious in young learners. Fourthly, the software design (i.e., relevant font size and its type, sufficiently big icons and control buttons, appealing interface, easy navigation, and the customization of the visual side of the application) reflects age-related issues, such as worsening eyesight or hearing. However, the novelty of this smartphone application consists in its joint development by the programmer (responsible for its operation and functionalities), the teacher (responsible for its academic content), and especially the end-user, the senior, whose social, cognitive, and technical needs have to be met. In this sense, the newly developed educational application seems to be irreplaceable and novel.

The limitation of this study might consist in a less direct usability testing since the testers were providing answers via e-mails, as well as in quite a homogeneous research sample, i.e., only the seniors from U3V were included. However, despite these limitations, the study has generated conclusive evidence, especially about the pedagogical and technical aspects of the well-designed educational smartphone application for learning English by cognitively unimpaired seniors.

## 6. Conclusions

The findings indicate that research of the use of smartphones and its applications for educational purposes by seniors is very rare. In addition, there is no suitable mobile application for foreign language learning by cognitively healthy older adults. Based on the literature findings and authors’ experience with programming and teaching English as a foreign language, a model pilot mobile application for learning English by cognitively unimpaired seniors has been developed and tested.

The findings confirm that the newly developed mobile application for teaching English meets the needs of cognitively healthy seniors, and it is acceptable and feasible. In addition, it indicates what technical (e.g., visual interface or easy navigation) and pedagogical (e.g., an instructional manual or adjusting to seniors’ learning pace or clear instructions) aspects should be strictly followed.

Furthermore, this pilot study aimed to show seniors that modern technologies are not just computer games or portals spreading disinformation. Mobile phones/smartphones and online teaching and learning can be of great benefit to seniors’ mental health. It can enable them to meet their peers, use their free time for productive self-study and thus expand, for example, their opportunities for travelling. This study also opens new chances for foreign language teaching and learning at the University of the Third Age since the courses do not have to be limited only to face-to-face classes at the university, but can also offer seniors further practice from the comfort of their homes.

Further research should include more empirical studies aimed at the exploration of educational mobile applications for older groups with respect to meeting their individual needs in order to enhance their overall well-being.

## Figures and Tables

**Figure 1 ijerph-18-06601-f001:**
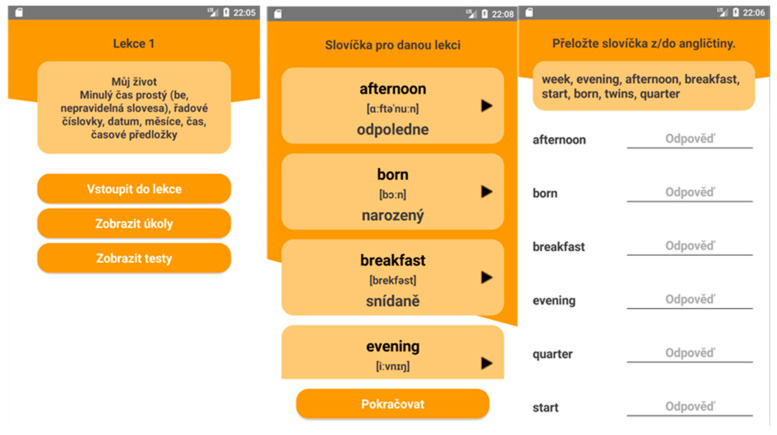
Illustration of Lesson 1 and vocabulary module.

**Figure 2 ijerph-18-06601-f002:**
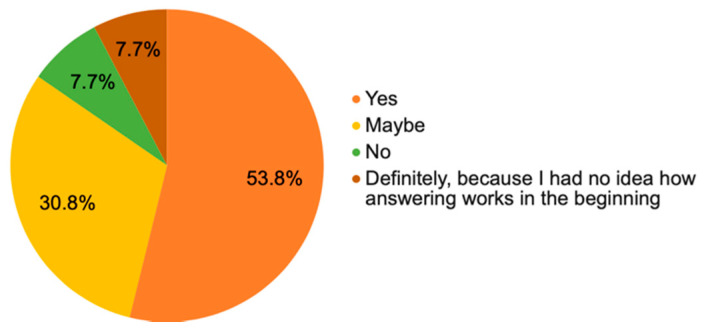
The importance of a user manual or a video tutorial.

**Figure 3 ijerph-18-06601-f003:**
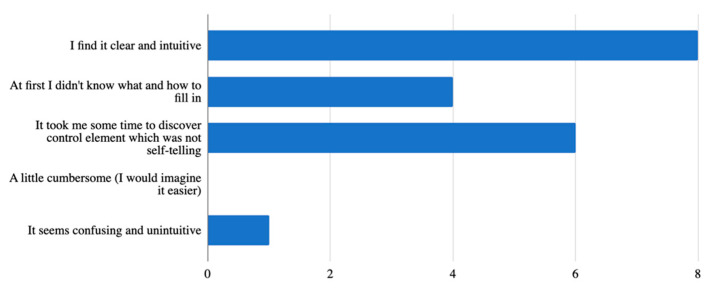
Control of the mobile application.

**Figure 4 ijerph-18-06601-f004:**
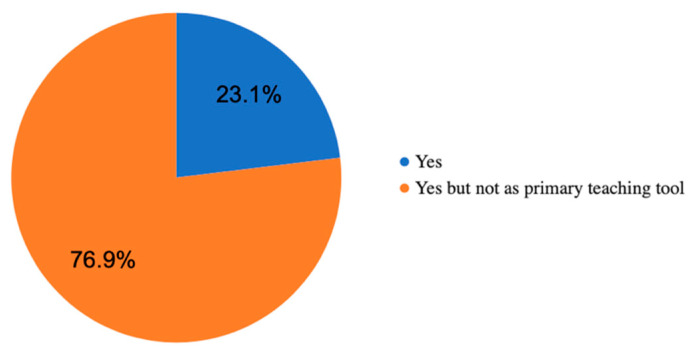
Use of the mobile application during their University of the Third Age (U3V) studies.

**Figure 5 ijerph-18-06601-f005:**
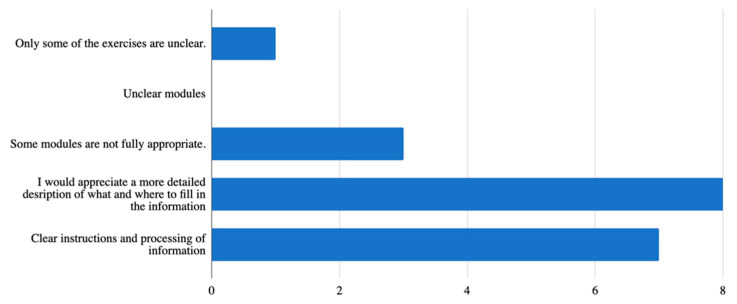
General opinion on the learning modules.

**Figure 6 ijerph-18-06601-f006:**
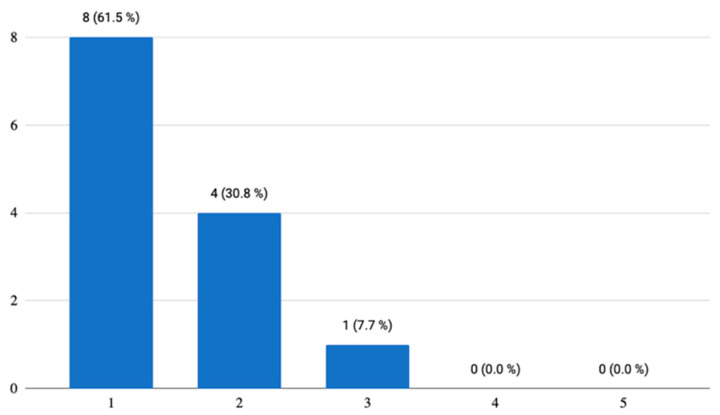
Use of the application during the U3V English courses.

**Figure 7 ijerph-18-06601-f007:**
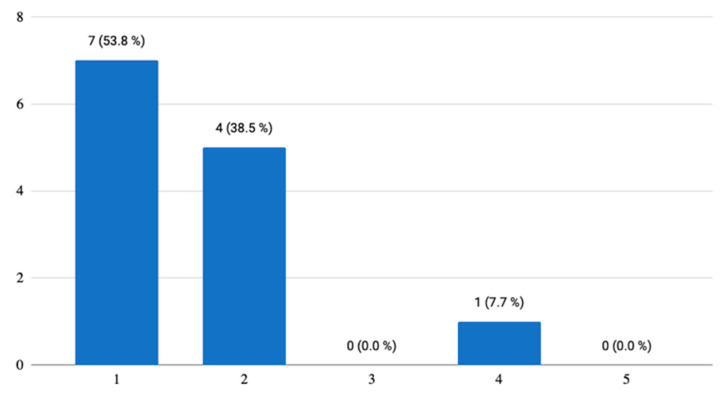
Seniors’ expectations of the mobile application.

**Figure 8 ijerph-18-06601-f008:**
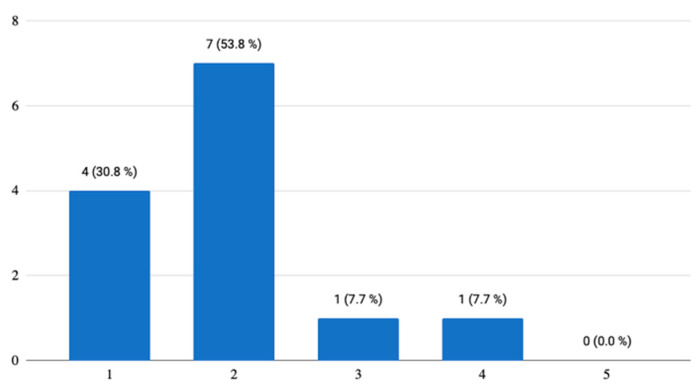
Control of the mobile application.

**Figure 9 ijerph-18-06601-f009:**
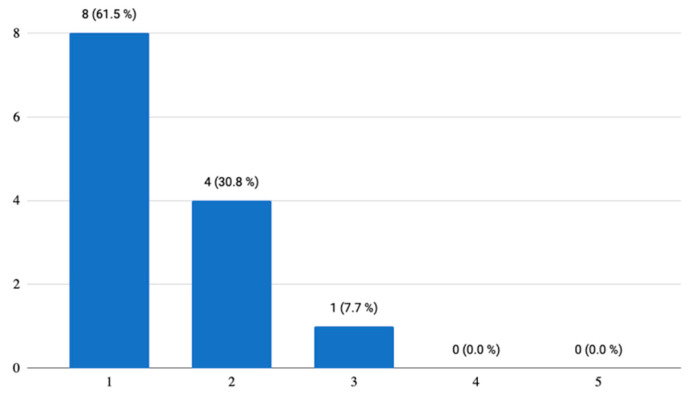
Overall evaluation of the learning modules, tasks and tests.

**Figure 10 ijerph-18-06601-f010:**
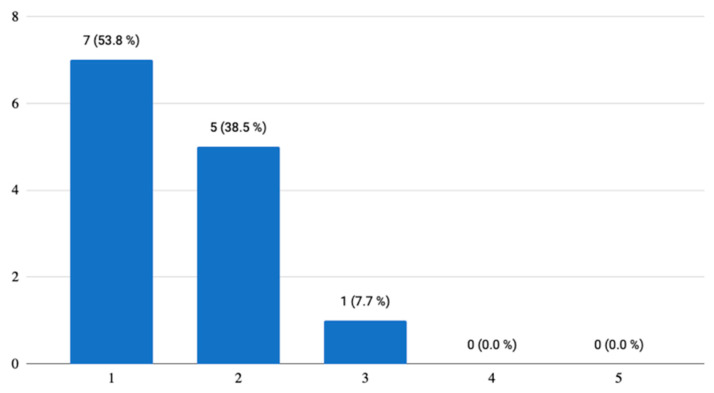
Overall evaluation of the mobile application.

**Figure 11 ijerph-18-06601-f011:**
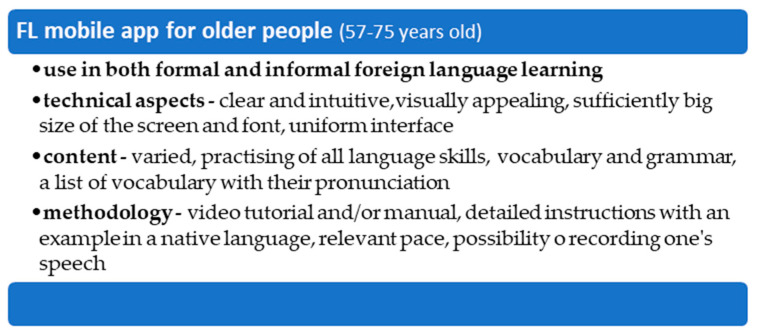
Summary of the key findings.

## Data Availability

The data are available upon request from the corresponding author.

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
