# Peer review of "A Novel Educational Smartphone Application for Cognitively Healthy Seniors: A Pilot Study"

_ijerph, 2021, doi:10.3390/ijerph18126601_

Round 1

Reviewer 1 Report

Thank you for the opportunity to read this research. The authors have addressed an important issue of the exploration of educational mobile applications for senior generation groups with respect to meeting their individual needs in order to enhance their overall well-being

  • Please add the reference citation of On average, older adults spend around 27 hours online per week.
  • The content was too repetitive, therefore I suggest reducing these sentences of p2-3, line 81-100. 
  • Although this was a pilot study, and the authors claim “Although the sample seems to be small, the usability testing requires a minimum of five users only”, but I still think the sample size (n=13) might be too small and the range of the age (55+ years with a mean age of 67 years) may not be suitable for this article's title of “Active Aging”. 
  • p5, line 186, “However, as it has been already said, only 76.47% (n = 13) of the addressed seniors tested and completed the questionnaire.” How about the attrition rate of this study? The author states the sample size was 13 in the methodology session, but it seems they included over 13 but some of the participants dropped out? Please explain the procedure of that.
  • I think this study didn’t use “two research methods”, but had two assessment tools. 

Author Response

Dear Reviewer,

Please see the attached file about our responses, as well as the revised manuscript.

Regards,

Authors

Reviewer 2 Report

The study is very interesting, it requires a small modification in the methodology

Author Response

(The authors gave the same response as above.)

Reviewer 3 Report

The introduction is too long and difficult to follow. The authors first stated that there is a decline in usage with age, but go on to about the increasing willingness in usage in adults as young as 55+. It is not clear which age group is the authors focusing on. Indeed, same problem is found in literature review, which is not linking well with the study objective.

The manuscript is simply describing an App development. More importantly, the mobile app was tested only in 13 people from 57 to 75 years of age. Data on samples' education level, knowledge of technology usage and cognitive functioning are lacking. 

Author Response

(The authors gave the same response as above.)

Reviewer 4 Report

Authors propose a novel educational application for elderly people. The aim is to teach english. The topic is interesting, however authors should:

  • highlight the novelty from the pedagogical point of view.
  • better describe the app
  • detail the usability test

Indeed, since authors claim this app is different from the others available, and it is more suitable for aging people, they should clearly describe which are the differences from the software design point of view (e.g. bigger text, etc.), but also from the pedagogical point of view. Are there differences in the way younger people learn? Is the leaning material organized in a way that facilitates older people, or it would be organized in the same way if the users were young people?

The app has been briefly described, but it is the main topic of this article. It should be better detailed together with the usability test. Authors should detail all the questions, group them by topic (as they did) and then discuss each group of questions.

Some minor comments:

  • Images are blurry, please use high quality images
  • literature review:
    • please add references/links for the mentioned apps
    • when you discuss the use of new technology by elderly people, you should specify where the research has been conducted. Indeed, the diffusion of the new technologies and their use is different in different countries.
  • Experimental settings: please add a summary of the experimental subjects, you have only specified the age range, it would be useful to know how many of them are 55 years old for example.

Author Response

(The authors gave the same response as above.)

Round 2

Reviewer 3 Report

The author has fully addressed the concerns raised previously.

Reviewer 4 Report

Authors have modified the article according with the reviewers' comments.